# Loss of Nup210 results in muscle repair delays and age-associated alterations in muscle integrity

Stephen Sakuma[1],[*] , Ethan YS Zhu[1],[*], Marcela Raices[1], Pan Zhang[2], Rabi Murad[2], Maximiliano A D'Angelo[1]

**Nuclear pore complexes, the channels connecting the nucleus with the cytoplasm, are built by multiple copies of ~30 proteins called nucleoporins. Recent evidence has exposed that nucleoporins can play cell type-specific functions. Despite novel discoveries into the cellular functions of nucleoporins, their role in the regulation of mammalian tissue physiology remains mostly unexplored because of a limited number of nucleoporin mouse models. Here we show that ablation of Nup210/Gp210, a nucleoporin previously identified to play a role in myoblast differentiation and Zebrafish muscle maturation, is dispensable for skeletal muscle formation and growth in mice. We found that although primary satellite cells from *Nup210* knockout mice can differentiate, these animals show delayed muscle repair after injury. Moreover, *Nup210* knockout mice display an increased percentage of centrally nucleated fibers and abnormal fiber type distribution as they age. Muscle function experiments also exposed that Nup210 is required for muscle endurance during voluntary running. Our findings indicate that in mammals, Nup210 is important for the maintenance of skeletal muscle integrity and for proper muscle function providing novel insights into the in vivo roles of nuclear pore complex components.**

## Introduction

Nuclear pore complexes (NPCs), the channels that connect the nucleus with the cytoplasm, have recently emerged as critical regulators of cell function and tissue homeostasis (1, 2). NPCs are large multiprotein structures composed of ~30 different proteins known as nucleoporins (3, 4). The expression of some nucleoporins as well as the composition and stoichiometry of NPCs has been found to vary in different cell types indicating that these structures can be specialized to play specific cellular roles (4, 5, 6). These findings help to explain why mutations in these proteins often result in diseases with markedly tissue-specific phenotypes. Yet, the physiological functions for most nucleoporins remains poorly understood.

The transmembrane protein Nup210 (originally known as Gp210) was the first nucleoporin identified and the first nuclear pore

component described to have tissue-specific expression (7, 8, 9). In previous work, we identified that the levels of Nup210 are differentially regulated during the differentiation of the C2C12 myoblast cell line (6). We found that although C2C12 myoblasts do not express Nup210, this nucleoporin is strongly up-regulated during myogenic differentiation. Preventing Nup210 expression in this model system strongly inhibited the formation of differentiated myotubes, whereas increasing its levels had the opposite effect (6). In differentiated C2C12 myotubes, Nup210 was found to regulate the activity of muscle genes through its association with Mef2C, a critical regulator of skeletal and cardiac muscle physiology; and to play a key role in the maintenance of nuclear envelope/ER homeostasis (10, 11). Consistent with a role in the regulation of skeletal muscle physiology, depletion of Nup210 in Zebrafish was found to alter the growth and maturation of differentiated muscle fibers and to result in progressive alterations in skeletal muscle integrity (10). But how Nup210 affects the development and function of skeletal muscle in a mammalian organism has not been investigated.

In this work, we used different *Nup210* knockout mouse lines to discover that Nup210 is dispensable for muscle development and for the differentiation of adult satellite cells in vitro and in vivo. We also provide evidence that Nup210-deficient animals have abnormalities in their response to muscle repair, show increased numbers of centrally nucleated fibers, develop alterations in fiber type distribution as they age, and show reduced muscle endurance. Our findings indicate that although Nup210 is not essential for the formation and growth of skeletal muscle in mammals, it is required for the proper maintenance of muscle integrity and function. Overall, our findings indicate that Nup210 is required for efficient muscle regeneration and highlight a role for this nucleoporin in the maintenance of mammalian skeletal muscle integrity with age.

## Results

### Nup210 ablation is dispensable for mammalian muscle development and satellite cell function

To study the role of the Nup210 nucleoporin in mammalian skeletal muscle development we took advantage of a constitutive *Nup210*

---

[1]Cell and Molecular Biology of Cancer Program, National Cancer Institute (NCI)-designated Cancer Center, Sanford Burnham Prebys Medical Discovery Institute, La Jolla, CA, USA   [2]Bioinformatics Core, NCI-Designated Cancer Center, Sanford Burnham Prebys Medical Discovery Institute, La Jolla, CA, USA

Correspondence: mdangelo@sbpdiscovery.org
*Stephen Sakuma and Ethan YS Zhu contributed equally to this work.

knockout mouse line that we have recently generated (12). In these animals, deletion of exon 2 results in the total loss of Nup210 protein. Because the low expression levels of Nup210 in muscle tissue makes it difficult to detect by Western blot (Fig S1A), we further confirmed *Nup210* knockout in muscles using immuno-precipitation and immunofluorescence approaches (Fig 1A–C). We found that *Nup210⁻/⁻* mice were born at Mendelian rates, showed no variance in weight compared with controls, and no survival differences up to 24 mo (Fig S1B and C). The only defect identified so far in these animals is a decreased number of circulating CD4 T cells (12). To investigate how ablation of Nup210 affects skeletal muscle physiology in mice, we isolated quadricep (quad) muscles from animals of 6–8 wk of age, sectioned them, and stained them with hematoxylin and eosin (H&E), as well as with anti-Laminin, a marker of the basal membrane that allows quantification of the number and diameter of muscle fibers. We found that young *Nup210⁻/⁻* mice show no obvious differences in the size, structure, or cross-sectional area (CSA) of quad muscles compared with control mice (Figs 1D–F and S1D). The lack of muscle alterations in young *Nup210⁻/⁻* mice is consistent with previous findings in Zebrafish showing that this nucleoporin is not required during embryonic muscle development (10).

Our previous work identified that Nup210 is required for the differentiation of the C2C12 myogenic cell line (6). Down-regulation of Nup210 in C2C12 myoblasts strongly inhibits their ability to form myotubes in vitro (6). To determine if adult *Nup210⁻/⁻* muscle progenitors show alterations in their differentiation potential, we isolated satellite cells from muscles of young mice and grew them in differentiation media to induce the formation of myotubes as described by Tierney et al (14). Consistent with our previous findings using the C2C12 myogenic model, we found that Nup210 is expressed in differentiated myotubes but not in satellite cells/myoblasts (Fig 2A). But in contrast to Nup210-depleted C2C12 cells, satellite cells from *Nup210⁻/⁻* mice were able to form myotubes in vitro (Fig 2B). Moreover, *Nup210⁻/⁻* satellite cells showed no significant decrease in their ability to differentiate compared to muscle progenitors isolated from control mice (Fig 2C). Because Nup210 is constitutively knocked out in these animals, it is possible that compensation during embryonic development could mask myogenic differentiation defects in these animals. To test this possibility, we acutely inactivated Nup210 only in satellite cells before inducing differentiation. For this, we used a conditional gene targeting approach in which we crossed a satellite-cell specific Cre transgenic mouse line, Pax7-CreER^T2 (15), with mice carrying a floxed Nup210 allele (12). In the *Nup210^f/f*/Pax7-CreER^T2 line, Nup210 ablation in satellite cells is induced by tamoxifen treatment. Control (*Nup210^+/+*/Pax7-CreER^T2) and *Nup210^f/f*/Pax7-CreER^T2 mice were injected with tamoxifen (three times every other day) to knockout Nup210, and satellite cells were isolated 48 h after the last injection. Satellite cells were then cultured in differentiation conditions and their ability to form myotubes was quantified. Consistent with our findings using the constitutive knockout mice, satellite cells acutely depleted of Nup210 were able to efficiently form myotubes (Fig 2D). Nup210 staining after differentiation confirmed the tamoxifen-induced knockout of this nucleoporin (Fig 2E). These findings indicate that Nup210 is not required for muscle development or adult satellite cell differentiation.

### *Nup210⁻/⁻* mice show delayed muscle regeneration

Even though *Nup210⁻/⁻* mice show no muscle alterations in homeostatic conditions, it is possible that challenging the muscles of these animals could expose underlying defects. To test if *Nup210⁻/⁻* mice have defects in muscle regeneration, we performed BaCl₂-induced muscle injury in the tibialis anterior (TA) as previously described (14). 5 d after injury, muscles were isolated, sectioned and stained with anti-Laminin. The ability of control and *Nup210⁻/⁻* mice to repair muscle was determined by quantifying the number and diameter of myofibers. These experiments showed that Nup210 levels are higher in regenerating muscle fibers (Fig S2) and also exposed that although *Nup210⁻/⁻* mice are able to repair muscle, they show a delay in the regeneration process (evidenced by an increased number of smaller fibers) when compared with control animals (Fig 3A).

To further confirm the ability of adult *Nup210⁻/⁻* satellite cells to form myofibers after injury, control and *Nup210^f/f*/Pax7-CreER^T2 mice carrying the ROSA-Tomato marker were treated with tamoxifen to acutely knockout Nup210 before inducing muscle injury with BaCl₂. The ability of Cre-expressing satellite cells, marked by the tomato fluorescent protein, to form myofibers in vivo was analyzed 5 d after injury in TA muscle sections. As shown in Fig 3B, Cre-expressing Tomato-positive cells from control (*Nup210^+/+*/Pax7-CreER^T2) or *Nup210⁻/⁻* (*Nup210^f/f*/Pax7-CreER^T2) similarly contributed to muscle repair.

To investigate if continuous muscle challenge could exacerbate the repair defects of *Nup210⁻/⁻* mice, we subjected these animals to serial muscle injury. For this, TA muscles of control and *Nup210⁻/⁻* mice were injured with BaCl₂ three times sequentially. Muscle injuries were spaced by 25 d to allow full muscle recovery before the next injury round, and muscle regeneration was analyzed 25 d after the last injury by measuring CSA. Under these conditions, *Nup210⁻/⁻* mice also showed a delay in muscle recovery (Fig 3C). This difference in myofiber CSA was similar to that observed in animals subjected to single injury, indicating no progressive deterioration of satellite cell function with continuous challenge. Altogether, these findings indicate that the alterations in muscle repair observed in *Nup210* knockout animals do not result from a decreased myogenic potential of skeletal muscle satellite cells.

### *Nup210⁻/⁻* mice accumulate centrally nucleated fibers with age

Using the Zebrafish model system, we previously uncovered that although fish lacking Nup210 do not have alterations in embryonic muscle development, they show a progressive deterioration of skeletal muscle structure as they age (10). To investigate if *Nup210⁻/⁻* mice also develop muscle alterations as they age, we analyzed quad muscle sections from 6- to 9-mo-old animals. We found that at this age, *Nup210⁻/⁻* mice have a significant increase in the percentage of myofibers with centrally located nuclei compared with controls, with no changes in the total number of myofibers (Figs 4A and S3A). A similar increase was observed in TA muscles (Fig S3B). We found that the higher number of centrally nucleated fibers is also present in the muscles of Nup210^f/f/Acta1-Cre mice, in which Nup210 is specifically depleted in differentiated adult skeletal muscle cells, indicating that this is a consequence of

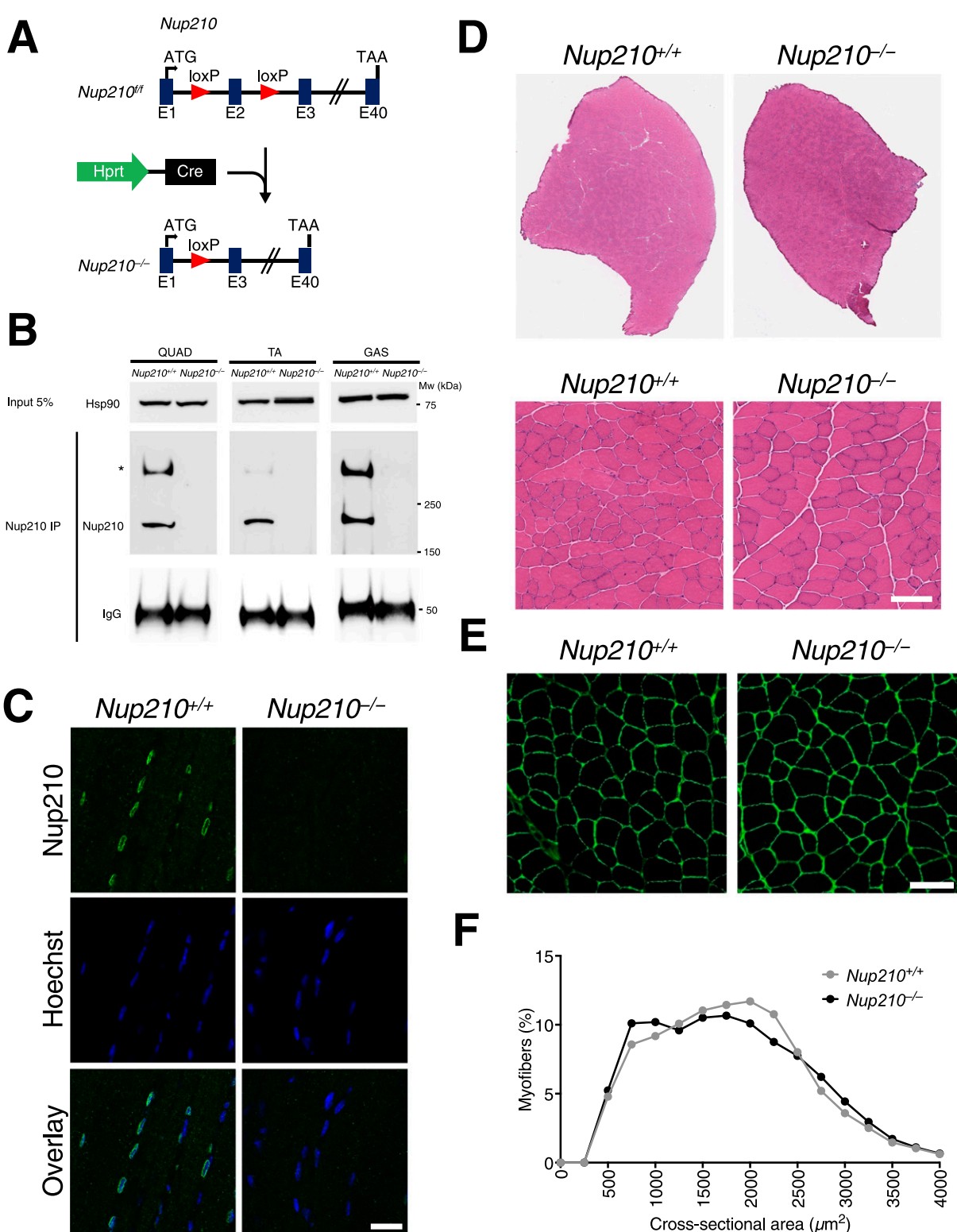

**Figure 1. Nup210 is dispensable for mouse development.**
**(A)** Schematic illustration of the generation of the *Nup210* knockout (*Nup210⁻/⁻*) mouse from the Hprt[Cre]-mediated recombination of the loxP flanked exon 2 of Nup210 (*Nup210^{f/f}*).
**(B)** Nup210 was immunoprecipitated from quadricep (Quad), tibialis anterior (TA), and gastrocnemius (GAS) muscles and the amount of protein in the immunoprecipitated was analyzed by Western blot. Hsp90 in the input is used as loading control for the immunoprecipitation and IgG shows the levels of anti-Nup210 used. Asterisk shows oligomeric forms of Nup210 previously described (13). **(C)** Representative images of Nup210 immunofluorescence staining in TA muscle longitudinal sections obtained from *Nup210⁺/⁺* and *Nup210⁻/⁻* mice. Scale bar, 25 μm. **(D)** Representative images of H&E full projections (top) and detailed views (bottom) of TA muscle transverse sections isolated from 6- to 8-wk-old *Nup210⁺/⁺* and *Nup210⁻/⁻* mice. Scale bar 100 μm. **(E)** Representative images of immunofluorescence staining for Laminin in quadricep muscles from 6- to 8-wk-old mice. Scale bar, 100 μm. **(F)** Quantification of myofiber cross-sectional area distribution in quadricep muscles from 6- to 8-wk-old mice (n = 3). Data are binned in 250 μm² bins and plotted as mean.

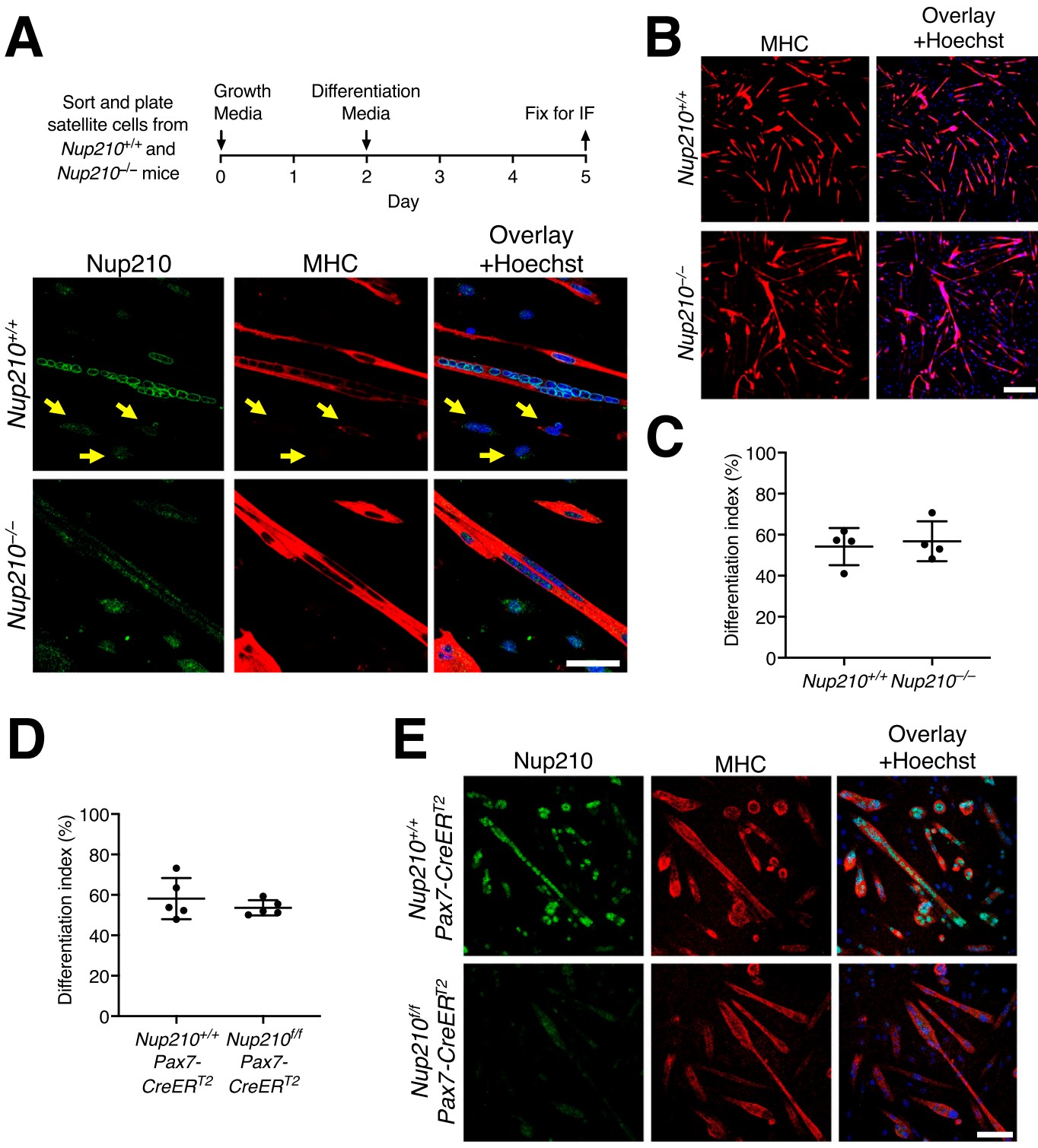

**Figure 2. Primary muscle satellite cells from *Nup210* knockout mice efficiently differentiate into myotubes.**
**(A)** Schematic illustration (top) and representative images immunofluorescence staining (bottom) of sorted satellite cells from $Nup210^{+/+}$ and $Nup210^{-/-}$ mice after induction of myogenic differentiation. Arrows indicate undifferentiated muscle progenitor cells which lack Nup210 expression. Scale bar, 50 $\mu$m. **(B)** Representative immunofluorescence images of differentiated satellite cells from $Nup210^{+/+}$ and $Nup210^{-/-}$ mice stained for myosin heavy chain (MHC) and Hoechst. Scale bar, 300 $\mu$m. **(C)** $Nup210^{+/+}$ and $Nup210^{-/-}$ satellite cells were sorted, differentiated, and stained for MHC. The percentage of nuclei in MHC-positive cells (differentiation index) was quantified. **(D)** $Nup210^{+/+}$/Pax7-CreER^{T2} and $Nup210^{f/f}$/Pax7-CreER^{T2} were treated with tamoxifen and satellite cells were isolated and differentiated in vitro. **(C)** The differentiation index (percentage of nuclei in MHC-positive cells) was quantified as in (C). **(D, E)** Representative immunofluorescence images of cells from (D) stained for Nup210 and MHC. Scale bar, 100 $\mu$m. Data are plotted as mean ± SD.

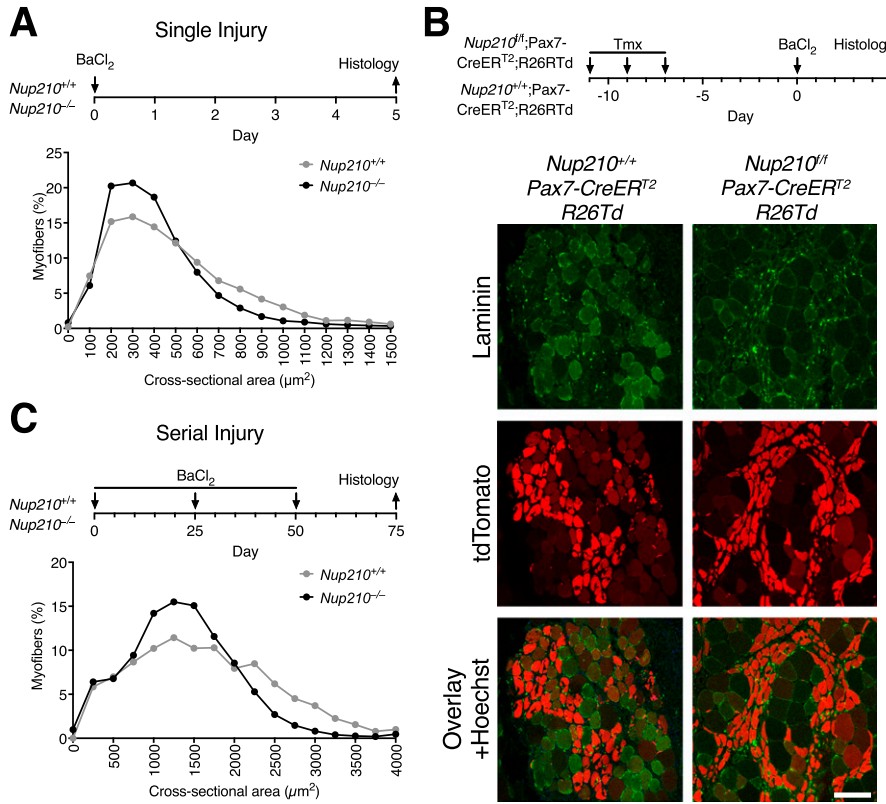

**Figure 3. Nup210 ablation results in delayed muscle regeneration.**
**(A)** Young (6–8 wk old) Nup210[+/+] and Nup210[−/−] mice were subjected to BaCl$_2$-induced muscle injury and muscle regeneration was analyzed 5 d later. Top: Schematic illustration of experimental approach. Bottom: Quantification of myofiber cross-sectional area distribution. n = 3, data are binned in 250 $\mu m^2$ bins and are plotted as mean. **(B)** Nup210[+/+]/Pax7-CreER[T2] and Nup210[f/f]/Pax7-CreER[T2] mice carrying a Cre-inducible tdTomato reporter (R26Td) were treated with tamoxifen before being subjected to BaCl$_2$ muscle injury. The tdTomato-positive myofibers were analyzed by immunofluorescence in muscle sections. Laminin was used as counterstain to detect muscle fibers. Top: Schematic representation of experimental approach. Bottom: Representative immunofluorescence images from TA sections. Scale bar, 100 $\mu m$. **(C)** Young Nup210[+/+] and Nup210[−/−] mice were subjected to BaCl$_2$-induced serial muscle injury (three injuries total), and muscle regeneration was analyzed 25 d after last injury. Top: Schematic illustration of experimental approach. Bottom: Quantification of myofiber cross-sectional area distribution. n = 3, data are binned in 250-$\mu m^2$ bins and are plotted as mean.

muscle-intrinsic alterations (Fig 4B). Like constitutive Nup210 knockout mice, these animals showed no significant differences fiber size distribution with control mice (Fig 4C and D). Our findings indicate that ablation of Nup210 is associated with an increase in the number of regenerating muscle fibers with no major alterations in muscle structure.

## Nup210 is required for the maintenance of normal skeletal muscle fiber type composition and for muscle endurance

Our previous work using the C2C12 myogenic model identified that Nup210 regulates the expression of muscle genes by modulating the activity of the Mef2C transcription factor (10). Mef2C is a critical

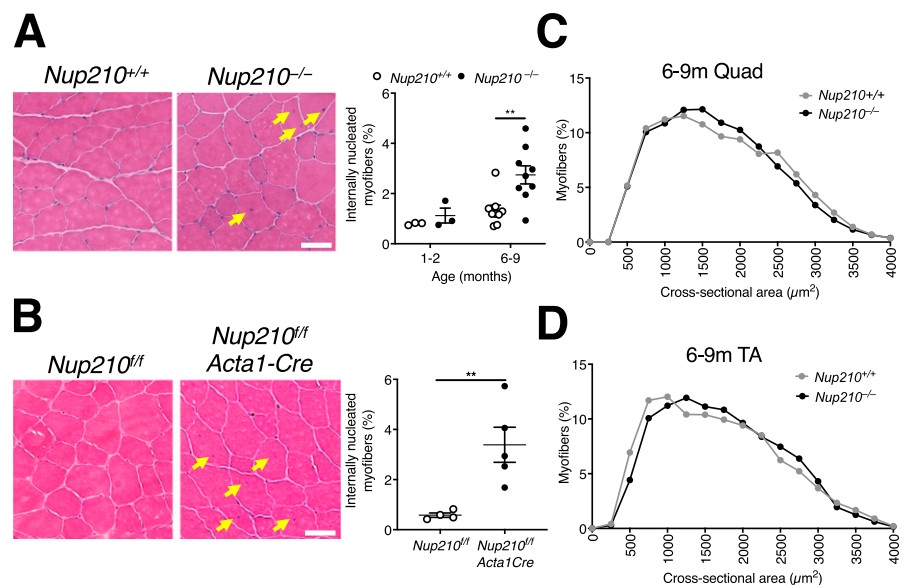

**Figure 4. Nup210 deficiency in muscle results in increased number of centrally nucleated myofibers.**
**(A)** Representative H&E images (left) and quantification of centrally nucleated myofibers (right) of quadricep muscle from 6- to 9-mo-old Nup210[+/+] and Nup210[−/−] mice (n = 3–9). Arrows denote centrally nucleated fibers. Scale bar 50 $\mu m$. **(B)** Representative H&E images (left) and whole section quantification of centrally nucleated myofibers (right) of quadricep muscle from 6- to 9-mo-old Nup210[f/f] and Nup210[f/f]/Acta1-Cre mice (n = 4–5). Arrows denote centrally nucleated fibers. Scale bar 50 $\mu m$. **(C)** Quantification of quadricep myofiber cross-sectional area isolated from 6- to 9-mo-old Nup210[+/+] and Nup210[−/−] mice (n = 3). Data are binned in 250 $\mu m^2$ bins. **(D)** Quantification of myofiber cross-sectional area distribution in TA isolated from 6- to 9-mo-old Nup210[+/+] and Nup210[−/−] mice (n = 4). **(A, B)** Data are binned in 250 $\mu m^2$ bins. **, P ≤ 0.01 by two-way ANOVA with Sidak's multiple comparisons test (A), and unpaired t tests in (B). Data are plotted as mean ± SEM in (A, B) and as mean in (C, D).

regulator of muscle fiber type composition that is preferentially active in slow oxidative type I fibers ([16], [17], [18]). Depleting Mef2C was found to reduce the number of type I fibers in mice, while increasing its expression levels has the opposite effect ([16], [19], [20]). To investigate if Nup210-deficient mice show alterations in fiber type composition, we performed immunofluorescence for fiber type–specific myosin isoforms (MyHC-I, MyHC-IIa, and MyHC-IIb) in sections of soleus, a muscle with a large fraction of type I fibers. We found that whereas young (6–8 wk old) animals show no significant differences in fiber type distribution (data not shown), 6- to 9-mo-old *Nup210* knockout mice have a significant decrease in the number of type I fibers and an increased number of type II fibers, particularly type IIx fibers, compared with control mice (Figs 5A and B and S4). The total number of fibers in the soleus of both mice was not different (Fig 5C). Further characterization of soleus muscles exposed an increase in fiber size (Fig 5D). The increased CSA was not restricted to a specific fiber type and was observed in both, type I and II fibers (Fig 5E and F), indicating a modest hypertrophy of the soleus muscle.

Alterations in muscle fiber type distribution can impact muscle performance. To determine if deletion of Nup210 affects skeletal muscle function, we performed treadmill assays and voluntary running experiments. Whereas no differences were observed in treadmill performance between control and *Nup210* knockout mice, ablation of Nup210 resulted in a trend to lower running distances in voluntary running experiments (Fig 5G and H). The reduced endurance of *Nup210* knockout mice could potentially result from the lower number of Type I slow muscle fibers, which are predominantly used during aerobic exercise and play a key role in long-duration muscle activity.

### *Nup210*$^{-/-}$ muscles show alterations in the expression of cell adhesion and immune genes

In the C2C12 myoblast differentiation system and the Zebrafish model, Nup210 was found to be required for the proper function of Mef2C, a key regulator of muscle development and fiber type distribution ([10]). To investigate if Mef2C function is affected in the muscle of *Nup210* knockouts animals, we performed whole genome expression studies by RNAseq. Numerous genes were deregulated upon Nup210 ablation (Fig 6A and Table S1), and Ingenuity Pathway Analysis (IPA) of genes changing expression levels by at least 1.5-fold (136 down-regulated, 30 up-regulated, q < 0.05) showed enrichment for cell adhesion/cytoskeletal and immune signaling pathways (Fig 6B). The alterations in cell adhesion/cytoskeletal genes observed in *Nup210* knockout muscles is consistent with previous studies ([6], [10], [21] Preprint), whereas the immune signature might be related with the increased number of regenerating fibers and with muscle damage. Notably, even though Mef2C was among the top predicted upstream transcriptional regulators altered by *Nup210* knockout in muscle (Fig 6C), only a few genes were found changed for this pathway (Table S2). The altered Mef2C-regulated genes in *Nup210* knockout muscle were mostly immune related, and we found that previously identified structural genes regulated by the Mef2C-Nup210 transcriptional complex in C2C12 myotubes were not affected in this tissue (Fig 6D). These findings suggest that Nup210 regulation of Mef2C activity in mammalian skeletal muscle

tissue might be compensated in the absence of this nucleoporin, which would explain the modest alterations in muscle integrity and function observed in *Nup210* knockout mice.

## Discussion

A significant body of evidence accumulated over the past few years indicates that nucleoporins, the components of the NPC channels, can play tissue-specific functions ([1], [2]). These findings have provided an explanation of why mutations in individual nucleoporins can result in diseases with markedly tissue-specific phenotypes ([22], [23]). Despite the growing number of specific functions recently described for nucleoporins, the in vivo roles of most NPC components in mammalian organisms have not been investigated. We previously identified that the transmembrane protein Nup210, the first described tissue-specific nucleoporin, is differentially expressed during myogenesis and is required for the differentiation of the myoblast cell line C2C12 ([6], [11]). We also identified that its depletion during Zebrafish development results in a progressive deterioration of skeletal muscle integrity and function ([10]). Here, we used a combination of constitutive and cell type-specific mouse knockout lines to investigate how deletion of Nup210 affects the physiology of mammalian skeletal muscle. We found that Nup210 ablation in mice does not alter embryonic muscle development or growth but leads to a modest increase in the number of centrally nucleated fibers as animals age. Centrally nucleated fibers are a common feature of muscles from patients and animal models of diverse muscle dystrophies ([24], [25], [26], [27], [28], [29]). They are also considered a marker of muscle regeneration, and generally result from increased muscle damage or from alterations in muscle repair. Analysis of muscle regeneration after injury showed that Nup210 deletion results in higher numbers of smaller fibers compared with control animals, suggesting a delay in the muscle repair process. Because muscle maintenance during adulthood requires the constant regeneration of muscle tissue, alterations in muscle repair could explain the accumulation of centrally nucleated fibers in the skeletal muscles of *Nup210* knockout animals as they age.

Muscle regeneration requires the activation and differentiation of satellite cells ([30]). Consistent with previous findings using the C2C12 myogenic model, we found that Nup210 is differentially expressed during satellite cell differentiation. Yet, ablation of this nucleoporin in primary satellite cells did not inhibit their ability to differentiate and form myotubes as it did in C2C12 cells. It is possible that the higher differentiation robustness of primary satellite cells when compared with C2C12 myoblasts can compensate for Nup210 defects. An alternative explanation is that C2C12 myoblasts may carry additional alterations that make them dependent on Nup210 function. In any case, our findings indicate that the decreased muscle repair observed in *Nup210* knockout animals is not a consequence of defects in muscle satellite cell differentiation.

Analysis of fiber type distribution in control and *Nup210* knockout mice showed that these animals also develop alterations in fiber type distribution, with a reduced number of type I fibers. How Nup210 contributes to the proper maintenance of fiber type

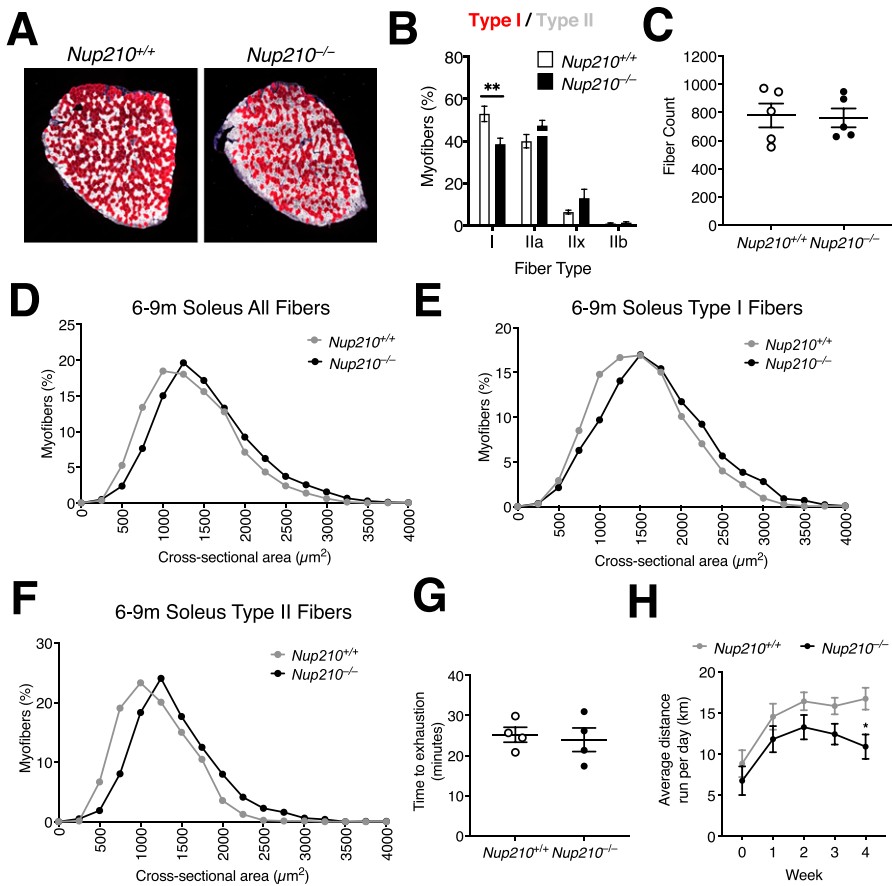

**Figure 5. Nup210 is required for the maintenance of normal skeletal muscle fiber type distribution.**
**(A)** Representative immunofluorescence images showing Type I (red) and II (white) fibers in soleus of 6- to 9-mo-old *Nup210*+/+ and *Nup210*−/− mice.
**(B)** Quantification of fiber type distribution in soleus of 6- to 9-mo-old *Nup210*+/+ and *Nup210*−/− mice (n = 5).
**(C)** Quantification of total fiber count for soleus muscles from 6- to 9-mo-old mice (n = 5). **(D, E, F)** Quantification of myofiber cross-sectional area for all fibers (D), Type I fibers (E), and Type II (IIa, IIx, and IIb) fibers (F) in soleus of 6- to 9-mo-old *Nup210*+/+ and *Nup210*−/− mice (n = 5). **(G)** *Nup210*+/+ and *Nup210*−/− mice were subjected to treadmill exercise and the time to exhaustion was quantified. **(H)** *Nup210*+/+ and *Nup210*−/− mice were subjected to 4 wk of voluntary running. The average distance ran per day each week was quantified and plotted. Data are plotted as mean ± SEM in (B), (C, G, H) and mean in (D, E, F). **P ≤ 0.01 by two-way ANOVA with Sidak's multiple comparisons test.

distribution in skeletal muscle is currently unknown. Our previous work identified that in differentiated C2C12 cells Nup210 regulates the activity of Mef2C, a transcription factor that is critical for fiber type specification and the formation of slow type I muscle fibers. But even when our RNAseq analyses of gene expression in *Nup210* knockout muscle identified Mef2C as deregulated transcriptional pathway, the altered genes identified for this pathway were immune-related and no significant changes in the activity of muscle genes previously identified to be co-regulated by Nup210-Mef2C were observed. This suggests that the function of Mef2C in *Nup210* knockout muscle might be compensated for in vivo. Notably, our previous work identified that Pom121, another trans-membrane nucleoporin, can partially compensate for Nup210-deficient muscle phenotypes by recruiting Mef2C to nuclear pores (10). In Zebrafish, co-depletion of Nup210 and Pom121 results in stronger muscle defects that cannot be rescued by increasing Mef2C activity. These findings suggest that Pom121 is not required for muscle function when Nup210 is present, but when Nup210 is absent, Pom121 might partially ameliorate the muscle defects by acting as an additional anchor for Mef2C. Unfortunately, because of the difficulty detecting Mef2C-NPC association in muscle sections by proximity ligation assays we have not yet been able to determine if Mef2C recruitment to nuclear pores is disrupted or maintained in *Nup210*−/− muscle. In future studies, it will be interesting to determine if Pom121 is indeed responsible for the compensation of Mef2C activity in *Nup210* knockout mice and to establish if other

transcription factors predicted to have altered activity in the absence of Nup210 contribute to the muscle defects observed in these animals.

## Materials and Methods

### Animals

Mice were fed ad libitum and housed under 12-h light/12-h dark cycles at the Sanford Burnham Prebys Medical Discovery Institute. For all mice, a marker-assisted speed congenic breeding strategy achieved 99.9% backcross to the C57BL/6J strain. Existing *Nup210*flox/flox (*Nup210*f/f) mice were crossed with HprtCre mice (stock no. 004302; The Jackson Laboratory) to generate *Nup210*−/− mice as previously described (12). *Nup210*f/f/Pax7-CreERT2 mice were generated by crossing Nup210f/f mice with Pax7-CreERT2 mice, obtained as a kind gift from A Sacco (15), and the crossing of *Nup210*f/f and HSA-Cre79, or ACTA1-Cre (stock no. 006149; The Jackson Laboratory) mice generated *Nup210*f/f/Acta1-Cre mice. *Nup210*f/f/Pax7-CreERT2 mice were further crossed with Gt(ROSA)26Sortm14(CAG-tdTomato)Hze mice carrying the R26RTd Cre reporter (31) to generate *Nup210*f/f/Pax7-CreERT2/Gt(ROSA)26Sortm14(CAG-tdTomato)Hze genotype mice. The Gt(ROSA)26Sortm14(CAG-tdTomato)Hze mice were kindly gifted by A Zovein.

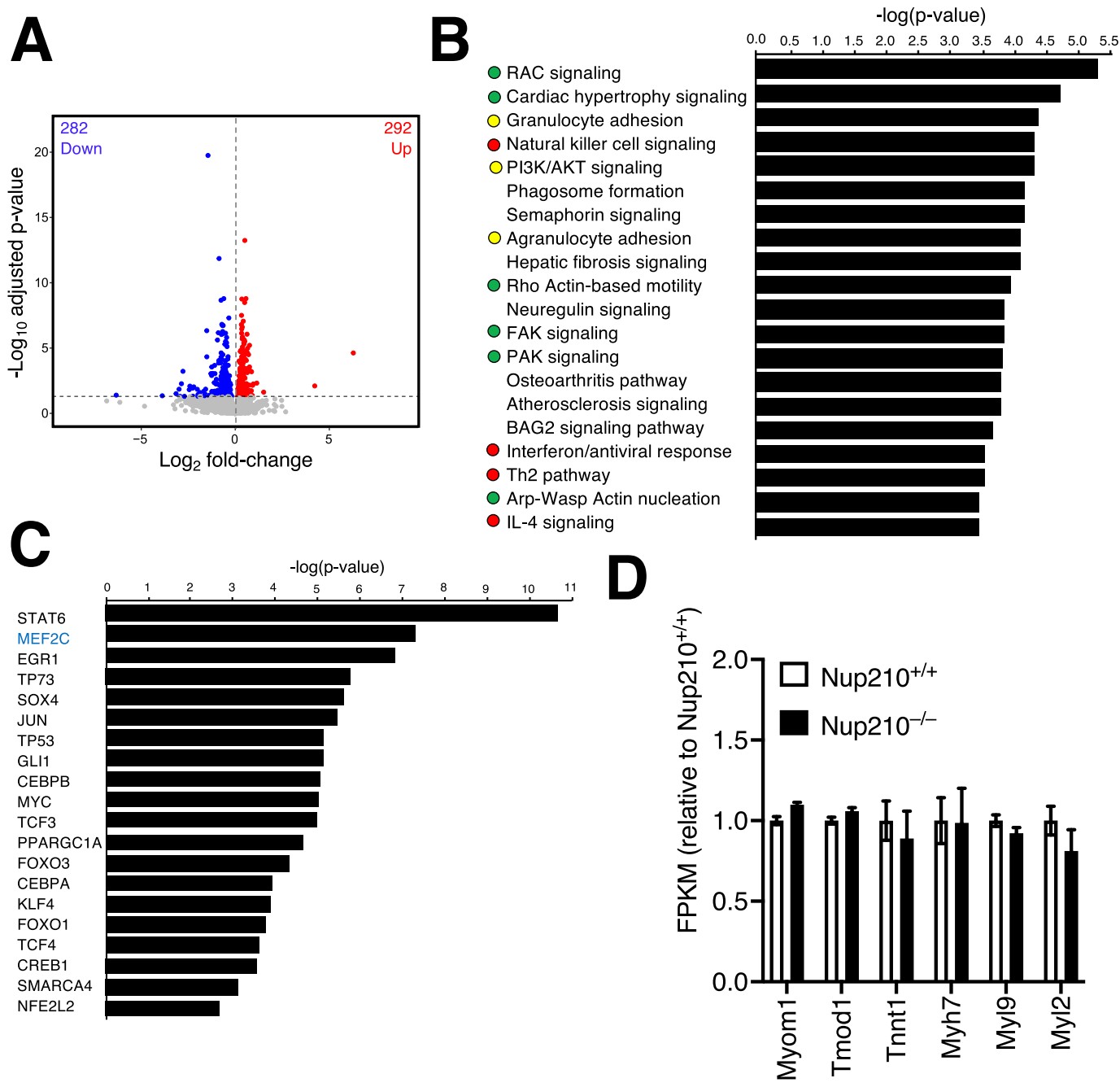

**Figure 6. Nup210 ablation affects the expression of cell adhesion and immune genes.**
**(A)** Volcano plot of RNA-seq transcriptome data for *Nup210* knockout quad muscle relative to control muscle. Significantly up- and down-regulated genes are reported as red and blue dots, respectively (q < 0.05). Genes not changing expression are represented as gray dots. **(B)** Ingenuity Pathway Analysis of genes deregulated in *Nup210* knockout quad muscle (>1.5 fold, q < 0.05). The top 20 pathways are shown. Green dots mark cell adhesion/cytoskeletal-related pathways, red dots show immune-related pathways, yellow dots show pathways associated with both. **(C)** Upstream transcriptional regulators predicted to be altered in *Nup210* knockout muscle by Ingenuity Pathway Analysis. Only pathways with at least 10 target genes affected were included in the analysis. The top 20 pathways are shown. Mef2C is shown in blue. **(D)** Expression levels for Mef2C muscle structural target genes from the RNAseq data from. Data are plotted as mean ± SEM.

## In vivo mouse experiments

All mouse experiments were reviewed and approved by the Institutional Animal Care and Use Committee and performed in accordance with institutional guidelines and regulations. For muscle injury, mice were fully anesthetized with isoflurane and the (TA) muscle was injured by injection of 50 µl 1.2% $BaCl_2$ solution with repeated piercing using a 27.5 G syringe. This procedure was repeated for serial injury experiments, with each injury spaced 25 d apart for a total of three injuries. 25 d after the final injury mice were euthanized and tissues were collected. For tamoxifen-inducible knockout experiments, *Nup210*[f/f]/Pax7-CreER[T2] mice were administered 2 mg

tamoxifen (T5648-1G; Sigma-Aldrich) suspended in corn oil via intraperitoneal injection three times with 48 h between injections.

## Mouse strength evaluation

For treadmill experiments, mice were acclimated to the Columbus Instruments treadmill for 15 min before turning the instrument on. The Simplex II controller was switched on and mice ran for 4 min at 11 m per minute. Treadmill speed was then increased by 1 m per minute each minute until the mice were exhausted. Time of exhaustion was recorded. Exhaustion was determined by failure to continue running for 5 s or stopping running three times for at least 2 s. For voluntary running experiments, mice were singly housed with a Med Associates wireless low profile running wheel (ENV-044). Running data were transmitted to a Med Associates USB interface hub (DIG-804).

## Satellite cell isolation and differentiation

Satellite cells were isolated as described previously (32). Briefly, TA, gastrocnemius and quadriceps muscles of mice were subjected to enzymatic dissociation (0.2% Collagenase type II and 0.02 units/ml Dispase II; Sigma-Aldrich) for 30 min. Muscle was then minced and filtered through a 70-$\mu$m nylon filter and incubated with the following biotinylated antibodies: CD45 (clone 30–F11), CD11b (Cat. no. #553309), CD31 (Cat. no. #5011513), and Sca1 (clone E13–161.7) (all BD Bioscience and 1:150 dilution). Streptavidin beads (1:10 dilution; Miltenyi Biotech) were then added to the cells together with the following antibodies: integrin-$\alpha$7–phycoerythrin (PE) (Cat. no. #R2F2; Ablab, 1:100 dilution) and CD34–Alexa 647 (clone RAM34; eBioscience, 1:50 dilution), after which magnetic depletion of biotin-positive cells was performed. The (CD45⁻CD11b⁻CD31⁻Sca1⁻) CD34⁺integrin-$\alpha$7⁺ satellite cell population was then live cell sorted by flow cytometry using a FACSAria II Cell Sorter (BD Biosciences) with a 70-$\mu$m nozzle. Sorted cells were plated in DMEM, 20% FBS, and GlutaMAX for 48 h. Cells were then differentiated in DMEM and 2% horse serum for 72 h before being fixed for immunofluorescence.

## Mouse muscle section preparations

Mice were euthanized by isoflurane overdose on the final day of the experiment. The indicated muscles were collected and frozen in OCT (Tissue-Tek) using isopentane equilibrated to liquid nitrogen. All muscles were cut into 10-$\mu$m frozen sections using a cryostat (Leica Biosystems). Frozen sections were thawed and processed for histology or immunofluorescence. Hematoxylin and eosin, Gomori trichrome, periodic acid–Schiff, and toluidine staining procedures were performed by the Sanford Burnham Prebys Histology Core, imaged using Leica Aperio AT2, and accessed using Aperio Scanscope software (Leica Biosystems).

## Western Blot analyses

Tissues were homogenized with RIPA buffer containing Protease and Phosphatase inhibitors (HALT Inhibitor Cocktail; Thermo Fisher Scientific) using the Qiagen TissueLyser and centrifuged at 3,000$g$ for 10 min at 4°C to remove tissue debris. Protein extracts were transferred to new tubes and DNA was sheared by passing them through a 29 G syringe at least 10 times. Extracts were centrifuged at 3,000$g$ for 10 min at 4°C to remove any insoluble fraction, and protein concentration was determined using the Pierce BCA reagent (Thermo Fisher Scientific).

For Western Blot analysis extracts were prepared at 2 mg/ml concentration with LDS Sample Buffer premixed and NUPAGE Sample Reducing Agent (Life Technologies), and incubated for 10 min at 70°C. 80–100 $\mu$g of protein were resolved by SDS–PAGE on NUPAGE Novex 3–8% tris-acetate protein gels (Life Technologies) and blotted to nitrocellulose membranes using an iBlot2 Dry Blotting System. Membranes were stained with Ponceau, washed with TBS, and blocked for 1 h at RT with TBS-0.1% Tween 20 (TBS-T) containing 5% non-fat milk, and incubated with primary antibodies overnight at 4°C. After three washes with TBS-T, the secondary antibody was added and incubated for 1 h at RT. Membranes were then washed and developed using SuperSignal West Pico Plus Chemiluminescent Substrate (Thermo Fisher Scientific).

For immunoprecipitations, 100 $\mu$l of tissue extract containing 2 mg of total protein were diluted to 1 ml in IP Buffer containing 50 mM Tris HCl, pH 7.4, 150 mM NaCl, 0.5% NP40, 1 mM EDTA, 1 mM MgCl$_2$, 0.5 mM DTT, and Protease and Phosphatase Inhibitors. Extracts were incubated with Protein A/G Dynabeads for 1 h to remove non-specific binding. 1 $\mu$g of anti-Nup210 antibody was added to each clarified extract. Immunoprecipitations were performed overnight at 4°C on a rotator. The next morning, Protein A/G Magnetic Beads (Pierce) were added, and samples were incubated on a rotator for 1 h at 4°C. Immunoprecipitations were washed three times for 5 min with IP buffer at 4°C and eluted from the beads by incubation with 100 $\mu$l 2× LDS sample buffer with a reducing agent for 15 min at 70°C. Five percent of the input and 60% of the volume of each immunoprecipitation were analyzed by Western blot as described above.

## Immunofluorescence

Frozen sections of mouse tissues were thawed and blocked in IF Buffer (0.2% Triton X-100, 10 mg/ml BSA, and 0.02% SDS in 1× PBS) at RT for 1 h, or overnight at 4°C. For fiber size analyses, sections were incubated with Rabbit anti-Laminin primary antibody in IF Buffer at RT for 1 h. For fiber typing, sections were additionally incubated with primary antibodies against MyHC-I, MyHC-IIa, and MyHC-IIb distributed by the Developmental Studies Hybridoma Bank (DSHB, University of Iowa) in IF Buffer overnight at 4°C. All sections were washed in IF buffer before incubation with secondary antibodies in IF buffer for 1 h at RT. All sections were washed in IF buffer, incubated with 1 $\mu$g/ml Hoechst 33342 (Life Technologies) for 5 min, mounted with VECTASHIELD (Vector Laboratories), and imaged via laser confocal microscopy (DMi8 Leica SP8). An ImageJ script (Supplemental Data 1) identified fiber boundaries using the anti-Laminin staining and measured CSA, and the fiber type was identified based on mean gray value of fluorescence measured for each of the fiber type-specific antibodies within the muscle fiber boundaries defined by the previously discussed script. The following antibodies were used: Nup210 (A301-795A; Bethyl Laboratories), MHC (DSHB, MF 20, deposited to the DSHB by Fischman DA), anti-Laminin (L9393; Sigma-Aldrich), anti-MyHC-I (DSHB, BA-D5, deposited to the DSHB by Schiaffino S), anti-MyHC-IIa (DSHB, SC-71, deposited to the DSHB by Schiaffino S), and anti-MyHC-IIb (DSHB, BF-F3, deposited to the DSHB by Schiaffino S).

### RNA sequencing

Quad muscles from four $Nup210^{+/+}$ and four $Nup210^{-/-}$ mice were snap frozen and then homogenized in Trizol reagent (Thermo Fisher Scientific) using the Qiagen TissueLyser LT. RNA was transferred to the aqueous phase by Trizol manufacturer specifications. After the addition of 70% ethanol, the total RNA was purified using the PureLink RNA Mini kit (Thermo Fisher Scientific). RNA quality and integrity was confirmed using the Agilent 2100 Bioanalyzer (Agilent Technologies). PolyA RNA was isolated using the NEBNext Poly(A) mRNA Magnetic Isolation Module and barcoded libraries were made using the NEBNext Ultra Directional RNA Library Prep Kit for Illumina (NEB). Libraries were pooled and single end sequenced (1X75) on the Illumina NextSeq 500 using the High output V2 kit (Illumina Inc.). Illumina Truseq adapter, polyA, and polyT sequences were trimmed with cutadapt v2.3 using parameters "cutadapt -j 4 -m 20 –interleaved -a AGATCGGAAGAGCACACGTCTGAACTCCAGTCAC -A AGATCGGAAGAGCGTCGTGTAGGGAAAGAGTGT Fastq1 Fastq2 | cutadapt –interleaved -j 4 -m 20 -a "A{100}" -A "A{100}" - | cutadapt -j 4 -m 20 -a "T{100}" -A "T{100}" -." Trimmed reads were aligned to mouse genome version 38 (mm10) using STAR aligner v2.7.0d_0221 (33) with parameters according to ENCODE long RNA-seq pipeline (https://github.com/ENCODE-DCC/long-rna-seq-pipeline). Gene expression levels were quantified using RSEM v1.3.1 (34). Ensembl gene annotations version 84 were used in the alignment and quantification steps. RNA sequence, alignment, and quantification quality was assessed using FastQC v0.11.5 (https://www.bioinformatics.babraham.ac.uk/projects/fastqc/) and MultiQC v1.8 (35). Biological replicate concordance was assessed using principal component analysis and pairwise Pearson correlation analysis. Lowly expressed genes were filtered out by applying the following criterion: estimated counts (from RSEM) ≥ number of samples × 5. Filtered estimated read counts from RSEM were compared using the R Bioconductor package DESeq2 v1.22.2 based on generalized linear model and negative binomial distribution (36). False discovery rate adjusted $P$-values were calculated using the Benjamini–Hochberg correction for multiple testing (Cufflinks). Differentially expressed genes were selected with a false discovery rate of q-value ≤ 0.05 and fold change ≥1.5 or ≤1.5. Differentially expressed genes were then analyzed using Ingenuity Pathway Analysis (IPA; QIAGEN).

### Data collection and analysis

Data were recorded and collected using Microsoft Excel. Figures were generated, and statistical analyses performed, in GraphPad Prism 8.4.2 (464). Statistical testing was conducted using two tailed unpaired $t$ test when comparing two groups, one-way ANOVA with Sidak's multiple comparisons test if significant differences were identified by one-way ANOVA for comparisons of three or more groups. Statistical significance was evaluated for $P < 0.05$. All immunofluorescence images were analyzed in Leica Application Suite X software v3.5.5.19976 and ImageJ v2.1.0/1.53c (National Institute of Health). All histology images were accessed using Aperio Scanscope software (Leica Biosystems). Live cell sorting was performed using the BD FACSDIVA Software (BD Biosciences). Voluntary running wheel data were acquired with Med Associates Wheel Manager (SOF-860).

## Data Availability

The muscle RNAseq data from this publication have been deposited to the Gene Expression Omnibus database (https://www.ncbi.nlm.nih.gov/geo/) and assigned the identifier GSE188803.

## Supplementary Information

## Acknowledgements

We thank Drs. Zovein and Sacco for kindly providing the R26RTd Cre reporter and Pax7-CreER[T2] mouse lines, respectively. MA D'Angelo was supported by a Pew Biomedical Science Scholar Award and Research Scholar Grant RSG-17-148-01-CCG from the American Cancer Society. This work was also supported by the Department of Defense (grant W81XWH-20-1-0212) and the National Institutes of Health (awards RO1AR065083 and RO1AR065083-S1). The content is solely the responsibility of the authors and does not necessarily represent the official views of the National Institutes of Health. This work was additionally supported by the National Cancer Institute (NCI) Cancer Center grant P30 CA030199, which supports the animal, flow cytometry, genomics, and bioinformatics cores at Sanford Burnham Prebys.

### Author Contributions

S Sakuma: conceptualization, data curation, formal analysis, validation, investigation, visualization, methodology, and writing—original draft, review, and editing.
E YS Zhu: data curation, software, formal analysis, and investigation.
M Raices: investigation and methodology.
P Zhang: resources and data curation.
R Murad: resources and data curation.
MA D'Angelo: conceptualization, data curation, formal analysis, supervision, funding acquisition, methodology, project administration, and writing—original draft, review, and editing.

### Conflict of Interest Statement

The authors declare that they have no conflict of interest.

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
