## [Reviewer comments · Life Science Alliance]

Life Science Alliance

Loss of Nup210 Results in Muscle Repair Delays and Age-Associated Alterations in Muscle Integrity

Maximiliano D'Angelo, Stephen Sakuma, Ethan Zhu, Marcela Raices, Pan Zhang, and Rabi Murad

DOI: <https://doi.org/10.26508/lsa.202101216>

Corresponding author(s): Maximiliano D'Angelo, Sanford Burnham Prebys Medical Discovery Institute

Review Timeline:

Submission Date:	2021-08-25
Editorial Decision:	2021-09-20
Revision Received:	2021-11-08
Editorial Decision:	2021-11-16
Revision Received:	2021-11-29
Accepted:	2021-11-30

Transaction Report:

September 20, 2021

Re: Life Science Alliance manuscript #LSA-2021-01216-T

Maximiliano A D'Angelo
Sanford Burnham Prebys Medical Discovery Institute
10901 North Torrey Pines Road
La Jolla, CA 92037

Dear Dr. D'Angelo,

Thank you for submitting your manuscript entitled "Loss of Nucleoporin Nup210 Results in Muscle Repair Delays and Age-Associated Alterations in Skeletal Muscle Integrity" to Life Science Alliance. The manuscript was assessed by expert reviewers, whose comments are appended to this letter. We invite you to submit a revised manuscript addressing the Reviewer comments.

Thank you for this interesting contribution to Life Science Alliance. We are looking forward to receiving your revised manuscript.

Sincerely,

B. MANUSCRIPT ORGANIZATION AND FORMATTING:

Reviewer #1 (Comments to the Authors (Required)):

This manuscript by Sakuma et al combine whole animal model of Nup210 gene knockout and constitutive cell type-specific mouse knockout lines to address how skeletal muscle is altered in the absence of the nucleoporin Nup210. The authors show that Nup210 deletion in mice does not compromise embryonic muscle development or growth but increases the number of centrally nucleated fibers as the animals age. The latter is characteristic of muscle dystrophies in addition to having a role in muscle regeneration, damage, and repair. The authors then showed a role for Nup210 in muscle regeneration after injury as Nup210 knockout induced increase in the numbers of smaller fibers. This finding indicates a delay in the muscle repair program. The authors argue that as muscle is continuously regenerating during adulthood, this defect in muscle repair may lead to the observed increase of centrally nucleated fibers in skeletal muscles as the mouse ages. Additionally, the authors showed abnormal fiber type distribution with a decrease in the number of type I fibers in Nup210 knockout mice. No clear phenotype related to the differentiation of satellite cells from Nup210 knockout mice was observed compared to control cells.

Overall, these are important findings that extend our understanding on the role of Nup210 in various skeletal muscle functions. The experiments are well done and the results are clear. There are a few points to be addressed as stated below:

Figure 1. A western blot showing lack of Nup210 in the knockout cells as control would make the figure clear and corroborate the image.

Figure 2. The cells marked with arrows in Figure 2A has a "green background". Depending on the cell type, deletion of an exon can generate other isoforms of a protein. The question here is whether these cells are expressing or not another isoform of Nup210. Please comment on this point.

Figure 5B. Is the difference in Iix statistically significant? It appears to be the case.

Reviewer #2 (Comments to the Authors (Required)):

1. A short summary of the paper, including description of the advance offered to the field.

Loss of the membrane spanning nucleoporin GP210/NUP210 reportedly blocks myogenesis in a C2C12-based tissue culture system, differentiation of mESCs into neuroprogenitors and myofiber growth, maturation and survival in zebrafish (as shown by the PI in previous publications). In this manuscript the authors show by NUP210 ablation in mice that the protein is dispensable for skeletal muscle formation and growth in this organism. Different to the C2C12-based model, satellite cells from KO-mice can differentiate in vitro and in vivo. Yet, the KO-mice show delayed muscle repair and at higher age an abnormal distribution of fiber types. Given the previously reported results, the KO-phenotype is surprisingly mild. Yet, this will be an important information for the field and stimulating the ongoing discussion about tissue-specific and disease-related functions of nucleoporins.

2. For each main point of the paper, please indicate if the data are strongly supportive. If not, explicitly state the additional experiments essential to support the claims made and the timeframe that these would require.

All the claims of the manuscript are well supported by the experiments. Different mouse-KO systems (constitutive and inducible Nup210 KO lines) are employed to validate the findings. As indicated, the comparatively weak muscle phenotype of the KO-mice is an important finding and should be published. In the current state the manuscript, however, does not provide an explanation for the striking phenotypic differences seen between the different test systems. One obvious explanation could be that the Mef2c complex, which is localized via NUP210 to nuclear pore complexes to control gene expression changes during muscle maturation in the zebrafish/human system (as previously reported by the PI's lab), is still present at nuclear pore complexes in the mouse KO-cells and fulfill its function in regulation of gene expression. Respective data (localization, gene expression) would certainly strengthen the manuscript and help the interpretation of the different phenotypes (time frame 3 month).

3. Lastly, indicate any additional issues you feel should be addressed (text changes, data presentation, statistics etc.).

Abstract: "a different mouse lines" - please correct.

Page 3: The correct citation for NUP210 showing tissue specific expression is, in my eyes, PMID 14697343. It would be also worth to mention at least once in the manuscript that NUP210 is also referred to as GP210 -its original and still widely used name.

Page 10: "Here, we used a combination of constitutive cell type-specific mouse knockout lines...". I guess an "and" is missing.

Page 10: "... to investigate for the first time": Given that other groups might also study NUP210 KO-mice this statement is hard to

evaluate.

Most, if not all error bars, represent SEMs. In my eyes the SDs would be more appropriate and useful to show. For Fig. 2, Suppl. Fig. 1 and Suppl. Fig. 3 it is not indicated whether error bars are SEM or SD.

Please check the description of the error bars in Fig 5: panel D and F do not show error bars but G and H.

Suppl. Fig. 3 lacks the label "A".

Response to reviewers

We would like to thank the reviewers for the thoughtful and positive evaluation of our discoveries, and for providing helpful suggestions that have allowed us to further improve our manuscript. As described below, we have addressed the reviewers' suggestions and added additional data to the manuscript (**Fig 1B, Fig 6, Fig S1A, Fig S4, Table S1, and Table S2**). Please find below the point-by-point response to the reviewers' comments.

Reviewer #1

Figure 1. A western blot showing lack of Nup210 in the knockout cells as control would make the figure clear and corroborate the image.

We entirely agree with the reviewer, and we thank him for the recommendation. We did not include a western blot image in our original submission because the levels of Nup210 in muscle are low and the western blot band for this protein is faint even when loading more than 100 μ g of protein in the gel. We have now included a western blot image in **Supplementary Figure 1**. Also, to overcome this low expression issues, we decided to immunoprecipitate Nup210 from different control and Nup210 knockout muscles and perform western blots on the immunoprecipitates. Now shown in **Figure 1B**, this approach results in strong Nup210 bands, including the reported oligomeric forms of the protein, and further confirms the full ablation of the protein in the muscles of Nup210 knockout mice.

Figure 2. The cells marked with arrows in Figure 2A has a "green background". Depending on the cell type, deletion of an exon can generate other isoforms of a protein. The question here is whether these cells are expressing or not another isoform of Nup210. Please comment on this point.

In our experience, the Nup210 antibody always gives a weak non-specific signal in immunofluorescence studies. When Nup210 is not present the background signal is stronger, likely due to the higher availability of antibody to recognize a non-specific antigen. We have seen this background noise in all cells lacking Nup210 including C2C12 myoblasts and stem cells **that do not express the protein** (D'Angelo et al, Dev Cell 2012, PMID: 22264802), as well as knockout hepatocytes, neuroprogenitors, T cells, and muscle cells (Borlido et al, Nat Immunol 2018, PMID: 29736031 & this work) (See some examples in **Reviewers' Figure 1**). The background signal is seen independently of the method used to eliminate Nup210 (shRNA, siRNA, CRISPr or flox-knockout) or the protein domain targeted for ablation. Because these methods reduce Nup210 levels using different mechanisms, it is unlikely that they all result in the production of the same alternative isoform. These findings strongly suggest that the signal observed results from antibody background rather than the creation of another isoform of the protein.

In addition, no evidence of an alternative isoform for Nup210 specifically expressed in knockout cells was found in our RNAseq experiments of muscle tissue or T lymphocytes. Exon 2 in the Nup210 gene was selected to knockout Nup210 because it is present in all identified splice variants and, thus, it is expected to eliminate all of them. Deletion of exon 2 generates an early frame shift on the coding sequence that results in a premature stop codon truncating the Nup210 protein from ~1880aa to <60aa. This small N-terminal peptide is not recognized by the C-terminal antibody used in this study. Although we cannot entirely discard the production of an unknown isoform for Nup210, altogether, our data strongly suggest that the signal more likely results from non-specific binding of the Nup210 antibody in the absence of the nucleoporin.

Figure 5B. Is the difference in IIX statistically significant? It appears to be the case.

The decrease in Type I fibers is statistically significant, as is the decrease in total Type II fibers (**now included as Supplementary Figure 4**). But when the type II fibers are divided into the different subtypes, the variability between animals makes the increase in Type IIa and IIX not significant.

Reviewer #2

One obvious explanation could be that the Mef2c complex, which is localized via NUP210 to nuclear pore complexes to control gene expression changes during muscle maturation in the zebrafish/human system (as previously reported by the PI's lab), is still present at nuclear pore complexes in the mouse KO-cells and fulfill its function in regulation of gene expression. Respective data (localization, gene expression) would certainly strengthen the manuscript and help the interpretation of the different phenotypes

This is a great point raised by the reviewer. In Zebrafish we previously identified that another transmembrane nucleoporin Pom121, can partially compensate Nup210 muscle phenotypes by recruiting Mef2C (Raices et al, Dev. Cell, 2017, PMID: 28586646). We found that co-depletion of Pom121 results in stronger muscle defects in this organism that cannot be rescued by increasing Mef2C activity. Our findings suggested that Pom121 may act as an additional anchor for Mef2C that is not required for muscle function when Nup210 is present, but that in its absence it might be sufficient to partially compensate its muscle defects. Thus, the reviewer's hypothesis that Mef2C activity might not be disrupted in these mice is a very valid one that we have also considered. As a first approach to test whether Mef2C activity is disrupted in these animals we performed immunofluorescence studies to determine whether the nuclear localization of the transcription factor is affected by Nup210 deletion; and proximity ligation assays (PLA) to determine whether its nuclear pore association is disrupted. Consistent with our previous findings, Nup210 knockout does not affect the levels of Mef2C in the nucleus (**Reviewer's**

Figure 2). Unfortunately, we have not been successful using PLA in muscle sections. PLA in tissues is difficult, and it is even more challenging for nuclear pore complex components. In our previous work, it was necessary to express tag versions of Nup210 and Mef2C to be able to detect the interaction *in situ*, which we cannot do in muscle. As an alternative approach to investigate whether Mef2C function is affected in the muscle of Nup210 knockouts we performed whole genome expression analyses by RNAseq. Analysis of altered pathways in Nup210 knockout muscle showed alterations in cell adhesion and immune signaling, consistent with muscle damage/repair. Even though analysis of upstream transcriptional regulators altered in Nup210 knockout muscle predicts alterations in Mef2C activity, most of the genes affected in this pathway are immune-related and the expression levels of Mef2C muscle target genes previously identified to be co-regulated with Nup210 show no difference between control and knockout muscles. These findings support the reviewer's hypothesis that Mef2C function is not disrupted in Nup210 knockout mice. This new data has been added in **Figure 6** and supplementary tables 1 and 2 and has been discussed in the manuscript.

3. Lastly, indicate any additional issues you feel should be addressed

Abstract: "a different mouse lines" - please correct.

Text has been corrected

Page 3: The correct citation for NUP210 showing tissue specific expression is, in my eyes, PMID 14697343.

We cited the Olsson et al, 1999 paper for Nup210 tissue-specific expression because it shows for the first time the differential expression of this nucleoporin in different embryonic tissues by *in situ* hybridization. This work precedes the 2004 paper from the same authors cited by the reviewer. But we agree with the reviewer, that the second paper focuses on the tissue-specificity of Nup210. We consider that citing both manuscripts would be the most appropriate approach and we have added the citation to the text. We hope the reviewer agrees with us.

It would be also worth to mention at least once in the manuscript that NUP210 is also referred to as GP210 -its original and still widely used name.

This is a great suggestion and has been addressed in the abstract and introduction.

Page 10: "Here, we used a combination of constitutive cell type-specific mouse knockout lines...". I guess an "and" is missing.

Text has been corrected

Page 10: "... to investigate for the first time": Given that other groups might also study NUP210 KO-mice this statement is hard to evaluate.

We agree with the reviewer and the statement has been removed

Most, if not all error bars, represent SEMs. In my eyes the SDs would be more appropriate and useful to show.

The use SD or SEM on scientific data remains highly debated. Our view on SEM/SD coincides with that of Tang, et al. J Pancreatol 2, 69–71 (2019), PMID: 34012702, and we favor SEM to represent most of our animal data (note that in experiments using in vitro cell culture we use SD to represent error, Figure 2). Our analyses are also based on the recommendation of our long-time collaborators and experts in the muscle field performing the same type of studies and who also prefer SEM for representation of error bars for these experiments (Tierney, M. T. et al. Nat. Med. 2014, PMID: 25194572)

For Fig. 2, Suppl. Fig. 1 and Suppl. Fig. 3 it is not indicated whether error bars are SEM or SD.

Figure legends have been corrected to include this information.

Please check the description of the error bars in Fig 5: panel D and F do not show error bars but G and H.

Figure legends have been corrected to include this information.

Suppl. Fig. 3 lacks the label "A".

Thank you pointing out this mistake. Suppl Figure 3 has been corrected.

[Figures removed by editorial staff per authors' request]

November 16, 2021

RE: Life Science Alliance Manuscript #LSA-2021-01216-TR

Dr. Maximiliano A D'Angelo
Sanford Burnham Prebys Medical Discovery Institute
10901 North Torrey Pines Road
La Jolla, CA 92037

Dear Dr. D'Angelo,

Thank you for submitting your revised manuscript entitled "Loss of Nup210 Results in Muscle Repair Delays and Age-Associated Alterations in Muscle Integrity". We would be happy to publish your paper in Life Science Alliance pending final revisions necessary to meet our formatting guidelines.

- please add the Twitter handle of your host institute/organization as well as your own or/and one of the authors in our system
- please note that titles in the system and manuscript file must match
- please consult our manuscript preparation guidelines <https://www.life-science-alliance.org/manuscript-prep> and make sure your manuscript sections are in the correct order
- please add your main, supplementary figure, and table legends to the main manuscript text after the references section
- please address Reviewer 3's comment about Figure S1
- please add scale bar to Figure S2 and indicate its size in Legend

A. FINAL FILES:

B. MANUSCRIPT ORGANIZATION AND FORMATTING:

**Submission of a paper that does not conform to Life Science Alliance guidelines will delay the acceptance of your

manuscript.**

The license to publish form must be signed before your manuscript can be sent to production. A link to the electronic license to publish form will be sent to the corresponding author only. Please take a moment to check your funder requirements.

Sincerely,

Reviewer #1 (Comments to the Authors (Required)):

The authors properly addressed my critiques. I recommend publication.

Reviewer #3 (Comments to the Authors (Required)):

The authors have addressed all points raised by the reviewers and the MS is in my eyes now acceptable for publication. CAVE: Figure S1, panel A needs a correction: both lanes are labeled Nup210+/+

November 30, 2021

RE: Life Science Alliance Manuscript #LSA-2021-01216-TRR

Dr. Maximiliano A D'Angelo
Sanford Burnham Prebys Medical Discovery Institute
10901 North Torrey Pines Road
La Jolla, CA 92037

Dear Dr. D'Angelo,

Thank you for submitting your Research Article entitled "Loss of Nup210 Results in Muscle Repair Delays and Age-Associated Alterations in Muscle Integrity". It is a pleasure to let you know that your manuscript is now accepted for publication in Life Science Alliance. Congratulations on this interesting work.

DISTRIBUTION OF MATERIALS:

Again, congratulations on a very nice paper. I hope you found the review process to be constructive and are pleased with how the manuscript was handled editorially. We look forward to future exciting submissions from your lab.

Sincerely,
